# Large scale sequence-based screen for recessive variants allows for identification and monitoring of rare deleterious variants in pigs

**Anne Boshove**[1]*, **Martijn F. L. Derks**[1,2], **Claudia A. Sevillano**[1], **Marcos S. Lopes**[1,3], **Maren van Son**[4], **Egbert F. Knol**[1], **Bert Dibbits**[2], **Barbara Harlizius**[1]

**1** Topigs Norsvin Research Center, 's-Hertogenbosch, the Netherlands, **2** Animal Breeding and Genomics, Wageningen University & Research, Wageningen, the Netherlands, **3** Topigs Norsvin, Curitiba, Brazil, **4** Norsvin SA, Hamar, Norway

* anne.boshove@topigsnorsvin.com

**Data Availability Statement:** For the 2Mb region around each significant QTL found in this study,

## Abstract

Most deleterious variants are recessive and segregate at relatively low frequency. Therefore, high sample sizes are required to identify these variants. In this study we report a large-scale sequence based genome-wide association study (GWAS) in pigs, with a total of 120,000 Large White and 80,000 Synthetic breed animals imputed to sequence using a reference population of approximately 1,100 whole genome sequenced pigs. We imputed over 20 million variants with high accuracies ($R^2$>0.9) even for low frequency variants (1–5% minor allele frequency). This sequence-based analysis revealed a total of 14 additive and 9 non-additive significant quantitative trait loci (QTLs) for growth rate and backfat thickness. With the non-additive (recessive) model, we identified a deleterious missense SNP in the *CDHR2* gene reducing growth rate and backfat in homozygous Large White animals. For the Synthetic breed, we revealed a QTL on chromosome 15 with a frameshift variant in the *OBSL1* gene. This QTL has a major impact on both growth rate and backfat, resembling human 3M-syndrome 2 which is related to the same gene. With the additive model, we confirmed known QTLs on chromosomes 1 and 5 for both breeds, including variants in the *MC4R* and *CCND2* genes. On chromosome 1, we disentangled a complex QTL region with multiple variants affecting both traits, harboring 4 independent QTLs in the span of 5 Mb. Together we present a large scale sequence-based association study that provides a key resource to scan for novel variants at high resolution for breeding and to further reduce the frequency of deleterious alleles at an early stage in the breeding program.

## Author summary

In this study we investigated the effect of over 20 million genetic variants on the growth rate and backfat thickness of approximately 140,000 pigs across two commercial breeds, with specific focus on recessive harmful variation. We identified 14 regions with a significant additive effect and 9 regions with a significant recessive effect on these traits. By looking at recessive effects we identified several rare deleterious variants with high impacts on

the following data is available for all animals: - Imputed sequence genotypes - Reference population genotypes - Imputation accuracies and allele frequencies - Raw GWAS output including p-values and effect sizes - Whole genome sequence statistics for the reference population All mentioned data is available on Open Science Framework (OSF): https://osf.io/fu6z5/.

**Funding:** The author(s) received no specific funding for this work.

**Competing interests:** All authors declare that the results are presented in full and present no conflict of interest.

animal fitness. These include a deletion on chromosome 15 in the *OBSL1* gene, which leads to a growth reduction of 100 grams a day on average. Interestingly, loss-of-function mutations in *OBSL1* are associated with short stature in humans. Looking at additive effects with this high-resolution dataset allowed us to gain more insight into the locus around the *MC4R* gene on chromosome 1. Here we found a small complex region containing several independent variants affecting both growth rate and backfat. With this study we have shown that by using several gene models and a large dataset, we can identify novel genetic variants at high resolution ($<0.01$ frequency) with significant impact on animal fitness and production. These results can help us in further eradicating deleterious genetic variants from pig populations.

## Introduction

In livestock, genomic selection has accelerated genetic gain due to its major impact on increasing the accuracy of breeding value estimation at a young age and reducing the generation interval [1]. Genomic selection, while boosting desirable trait improvement, can inadvertently exacerbate the frequency of deleterious alleles. By favoring individuals with superior trait-associated genetic markers, carriers of rare recessive deleterious alleles can be unknowingly propagated in breeding populations, potentially leading to the expression of harmful phenotypes in subsequent generations and compromising overall genetic health [2]. Balancing accelerated genetic gain with the need to mitigate the accumulation of deleterious alleles becomes a critical consideration in sustainable breeding strategies.

Deleterious recessive alleles can remain hidden because the harmful effects are only present in a homozygous state, and their impact may not be fully captured by traditional additive genetic models [2]. Especially for low frequency alleles, a large study population is crucial for identifying deleterious recessive alleles because the number of homozygous animals for the minor allele is small. In addition, imputing to sequence is essential for identifying deleterious alleles as it extends the scope of genetic analysis beyond genotyped variants, enabling the discovery of rare and non-genotyped variants associated with deleterious effects [3]. Imputation is a method that allows for predicting the genotypes of organisms at a higher density, based on a reference population of which this higher density data is already available [4]. In commercial pig populations, imputation with good accuracies is possible because there is a limited set of haplotypes segregating [5]. By performing imputation, we accurately predict the large majority of genetic variation within populations as long as a sufficiently large reference population is available. With whole genome sequencing (WGS) becoming more accessible and affordable, it is now possible to obtain reference populations allowing for performing imputation up to whole genome sequence level with high accuracies. Imputation to sequence is not only useful to fine map QTL regions, but also to identify novel deleterious alleles affecting the fitness of animals by focusing on non-additive effects. This can be done by performing genome-wide association studies (GWAS) using different models [6]. Previous studies have identified non-additive (recessive) effects, mainly by focusing on depletion of homozygotes [7,8]. This method allows for identifying recessive variants with lethal effects, but not recessive variants decreasing fitness without leading to death.

By performing GWAS, we can test the association between SNPs and phenotypic records of traits of interest, which allows us to identify quantitative trait loci (QTLs). Several models can be used for GWAS, with the additive model being most commonly used [9]. An additive GWAS model assumes that the effects of different alleles on the trait are cumulative and can be

estimated linearly based on the number of alleles present. However, this model can fail to pick up non-additive genetic variation, such as recessive and dominance effects. Recessive effects are especially of interest when trying to identify deleterious variants [2]. To identify SNPs with recessive effects, we can use a non-additive model. Recently several novel deleterious (coding) variants were identified in cattle using a non-additive GWAS on imputed sequence of >100,000 individuals [10].

In this study, we performed imputation to sequence of 120,000 pigs of a Large White sow breed and 80,000 pigs of a Synthetic boar breed. We performed GWAS on this imputed sequence data using both an additive and a non-additive (recessive) model for the production traits growth rate (GR) and backfat (BF). Using the non-additive model, we identified novel low frequency deleterious alleles affecting our traits of interest including loss-of-function mutations that can be purged from the population.

## Results

### Imputation to sequence

The dataset consists of 120,147 Large White and 81,250 Synthetic animals genotyped on medium density SNP panels (25K/50K). These animals were first imputed to 660K density with a reference population of 3500 animals. Subsequently, the imputed 660K genotypes were imputed to whole genome sequence level, with a reference population of 1069 animals (**S1 Fig**). We filtered the results to only include variants with an allele count of at least 100. This gave us a total of 28,190,307 variants for Large White and 24,124,813 variants for Synthetic. The majority of these variants were SNPs (**Table 1**). For SNPs with MAFs above 1–2% we were able to obtain very good imputation accuracies ($R^2 > 0.9$), and even for half of the SNPs below 1% frequency we obtained accuracies of $R^2 > 0.5$ (**Fig 1**).

### Additive and non-additive sequence-based GWAS on growth rate and backfat

We performed both an additive and non-additive (recessive) GWAS on growth rate and back-fat thickness in the Large White (**Fig 2A and 2B**) and Synthetic breed (**Fig 2C and 2D**) using the imputed sequence data. For Large White, we used 67K phenotypes for growth rate and 72K for backfat. For the Synthetic breed, we used 74K phenotypes for both traits. From the results we observe a large number of QTLs, with distinct QTLs for additive and non-additive effects. Across traits and breeds, we find a total of 14 additive and 9 non-additive QTLs, using a significance threshold of *p*-value < 1E-10 (**Fig 3**). For each QTL, we examined the top SNPs and their associated effects on genes using Ensembl VEP [11], SIFT scores [12] and pCADD scores [13], and we assessed the impact these QTLs have on the phenotypes (**Fig 4 and S1 Table**). We managed to identify potential causal variants for some of these effects.

**Table 1. Number of variants imputed to sequence split by variant type and MAF.**

|  | Large White | | Synthetic | |
| --- | --- | --- | --- | --- |
|  | SNPs | Indels | SNPs | Indels |
| **Total** | 22,840,678 | 5,349,629 | 19,398,280 | 4,726,533 |
| **< 1% MAF** | 4,329,777 | 600,075 | 4,496,085 | 641,335 |
| **< 5% MAF** | 8,109,989 | 1,472,040 | 7,510,302 | 1,400,929 |

MAF, minor allele frequency; SNP, single nucleotide polymorphism; Indel, insertion / deletion.

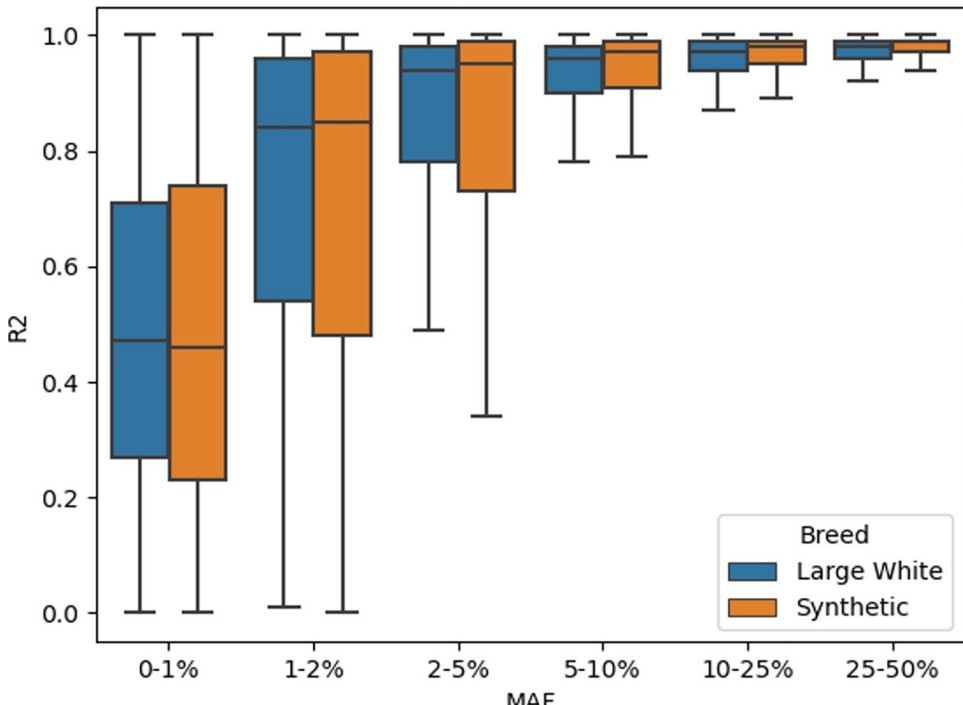

**Fig 1. Imputation accuracies (R2) of all SNPs grouped by Minor Allele Frequency (MAF).**

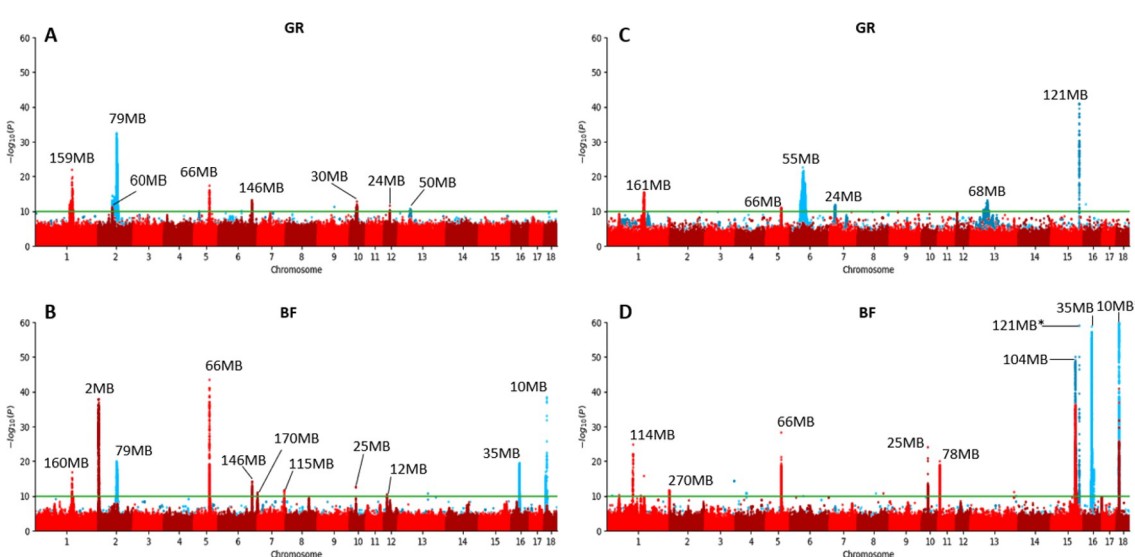

**Fig 2. Additive (red) and recessive (blue) Manhattan plots for growth rate (GR) and backfat (BF).** For peaks with a p-value significance below 1E-10 (green line) the genomic location is shown. QTLs only supported by a single significant SNP are neglected. A) Large White growth rate B) Large White backfat C) Synthetic growth rate D) Synthetic backfat. *The 121MB and 10MB peaks have top SNPs with p-values of 6,91E-154 and 1,63E-135 respectively.

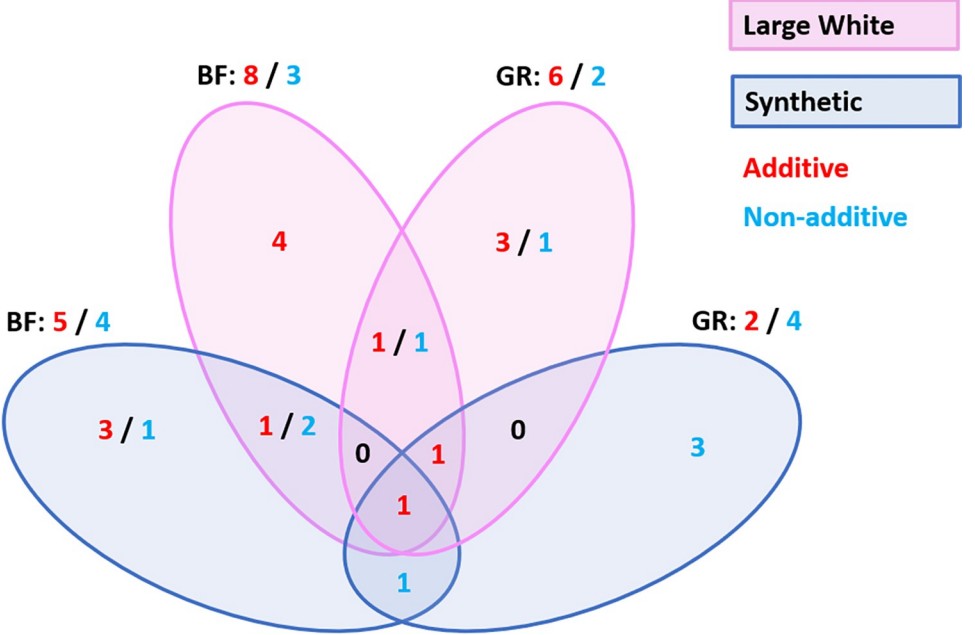

**Fig 3. Number of significant GWAS QTLs overlapping between breeds and traits.**

## Non-additive GWAS identifies loss-of-function variants associated with poor performance in homozygous individuals

**Non-additive effect on chromosome 2 shows decrease in growth rate and backfat in Large White.** We found a very significant non-additive QTL on chromosome 2 for both growth rate and backfat. The QTL is located around 79Mb with MAFs of 10%. The SNPs in this QTL show a negative impact on growth and backfat in homozygous individuals, and a negligible effect in heterozygous individuals. Homozygous animals grow on average 20 grams a day less (0.24 SDs) and have 0.7mm less backfat (0.47 SDs). The top SNP for growth is located at 79,261,674bp and the top SNP for backfat is located at 79,236,726bp. Both SNPs are also present very significantly in the other traits' GWAS. The QTLs consist mostly of intron variants of *ADAMTS2* (A Disintegrin-Like and Metalloproteinase with Thrombospondin Type 1 Motif 2). Interestingly, the QTL comprises a deleterious missense SNP (pCADD: 22.6, SIFT: 0) in the *CDHR2* (Cadherin-Related Family Member 2) gene at 81,336,954bp, known to affect body size in knockout mice [14].

**Stop gain SNP in *ANKRD55* affects backfat levels in both Large White and Synthetic breeds.** For backfat we find a very significant non-additive QTL on chromosome 16 that segregates in both breeds. The SNPs in the QTL have a MAF of around 10% in Large White and 19% in the Synthetic breed. The top SNP for the Synthetic breed and also a very significant SNP in the Large White GWAS is a stop gain SNP in the *ANKRD55* (ankyrin repeat domain 55) gene, located at 35,245,909bp. Homozygous animals show increased backfat levels, whereas heterozygous animals show some decrease in backfat. Homozygous animals have 0.29mm (0.17 SDs) more backfat in the Large White breed and 0.18mm (0.13 SDs) more backfat in the Synthetic breed. Heterozygous animals show 0.11mm less backfat in both breeds.

**Missense SNP in *MPIG6B* decreases growth rate in the Synthetic breed.** In the Synthetic breed, we observe a non-additive QTL on chromosome 7. Homozygous animals show a

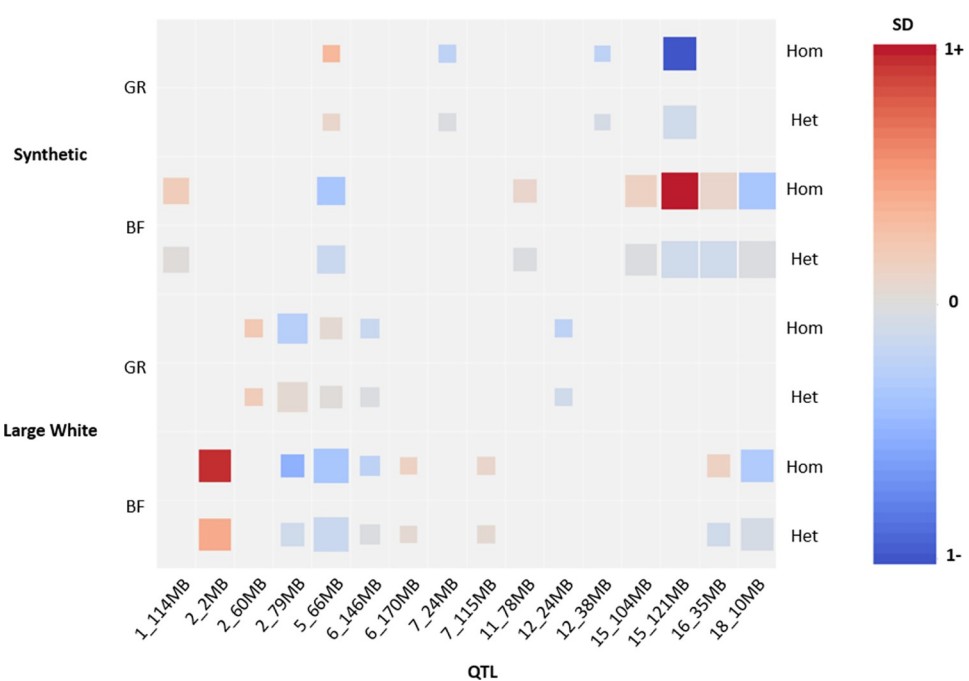

**Fig 4. Phenotypic effect sizes of top SNPs from the significant QTLs.** QTLs with phenotypic effects < 0.1 standard deviations are excluded. Square size indicates significance (bigger square is lower *p*-value, ranging from 1E-10 to 1E-50+).

decrease in growth of 30 grams a day (0.2 SDs). The fourth most significant SNP in this QTL is a deleterious missense SNP (pCADD: 24.4, SIFT: 0.01) in *MPIG6B (*Megakaryocyte And Platelet Inhibitory Receptor G6b). The SNP is located at 23,835,601bp with a MAF of 28%.

**Frameshift variant in *OBSL1* strongly affects growth rate and backfat in the Synthetic breed.** In the Synthetic breed, we identified a very significant QTL for both growth rate and backfat on chromosome 15. The top SNP is the same for both traits (121,500,039bp) and has a

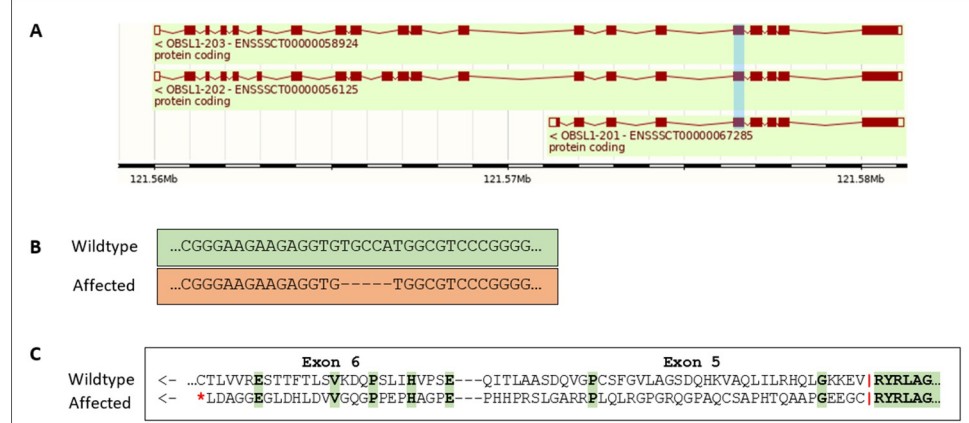

**Fig 5. Genomic overview of the *OBSL1* frameshift variant.** A) *OBSL1* gene model adapted from Ensembl 110 showing the genes three transcripts with exons (boxes) and introns (lines) The exon where the frameshift is located (exon 5) is highlighted in blue. B) DNA sequence alignment showing the 5bp deletion causing a frameshift in affected animals. C) Protein alignment showing the frameshift inducing a premature stop-codon on exon 6.

MAF of 5%. Homozygous animals show very poor growth and highly elevated levels of backfat, growing on average around 100 grams a day less (-1.08 SDs) and showing an increase in back-fat of 2.2mm (2.24 SDs) compared to non-homozygotes. We identified the most likely causal mutation to be a frameshift variant located at 121,576,506bp, in high LD with the top SNP from the GWAS ($R^2$ = 0.95). The frameshift is caused by a 5bp deletion in the 5th exon of the *OBSL1* (Obscurin Like Cytoskeletal Adaptor 1) gene (**Fig 5** [15]). This induces a premature stop-codon in the 6th exon. To validate the presence of the 5bp deletion we genotyped 31 pigs from three litters where the sow was carrier of the deletion (using a dye-labeled primer). We confirmed the presence of the deletion and confirmed 15 carriers using this test (**S2 Table**).

## Additive GWAS identifies known, novel, and complex QTLs with varying effects on performance

**Additive model shows expected QTLs on chromosomes 1 and 5 in Large White and Synthetic breed.**   In the additive GWAS results we find two QTLs that we consistently observe when performing GWAS on these traits. One is located on chromosome 1 where we find a missense SNP in the *MC4R* (melanocortin 4 receptor) gene located at 160,773,437bp (rs81219178). This missense SNP has been described before, animals with the major allele (G) show less backfat while animals with the minor allele (A) show faster growth [16], hence the QTL showing up for both traits. On chromosome 5, we find a QTL especially significant for backfat. The top SNP of this QTL is an intron variant of the *CCND2* (cyclin D2) gene affecting gene expression, located at 66,103,958bp (rs80985094). The G allele for this SNP has been reported to increase backfat [16].

**Low frequency additive QTL on chromosome 2 strongly affects backfat in Large White.**   At the start of chromosome 2 we find a very significant novel QTL for backfat, with SNPs located from 17Kb to 2.3Mb. These SNPs have very low frequencies with MAFs between 0.4%-0.5%. Despite the low frequency, the SNPs in the QTL have high imputation accuracies ($R^2$>0.9). Heterozygous animals on average have an increased backfat thickness of around 0.7mm (0.43 SDs). Due to the very low frequency of these variants, our dataset only includes 3 homozygous animals, of which 2 show an increase in backfat of over 2.3mm (1.36 SDs). The QTL is located in a region encompassing different genes including *NAD-SYN1* (NAD synthetase 1), *INS (*Insulin*)* and *IGF2* (Insulin like growth factor II). Both *INS* and *IGF2* play a role in glucose regulation and affect lipid metabolism, making them interesting candidate genes for the phenotypic effects we observe in this QTL. Another significant SNP present in the QTL is a missense SNP in *ANO9*, (anoctamin 9) located at 245,676bp with a pCADD score of 21.78.

**Small complex region on chromosome 1 contains several independent variants affecting growth rate and backfat.**   On chromosome 1 we find a very complex region located from approximately 150-165Mb (**Fig 6**). This region is present for both traits and both breeds, but most effects and highest significances are observed for growth rate in Large White. In the Synthetic breed we find only a single significant SNP for backfat (**Fig 6D**), which is also present in the other breed and trait.

In this region we find 5 independent QTLs within this span of 15Mb, of which 3 within a 3Mb span. Within some of the QTLs, we find SNPs with different and opposing effects. The majority of the top SNPs of the different QTLs and effects are not in linkage disequilibrium (LD) (**Table 2**), indicating that these are mostly independently segregating haplotypes.

The QTLs at 150 and 154Mb in Large White both consist of only intergenic variants with allele frequencies ranging from 30–35%. One of the positive effects we observe in the 3Mb span from 159-162Mb is due to a missense SNP in the *MC4R* gene. This effect is represented

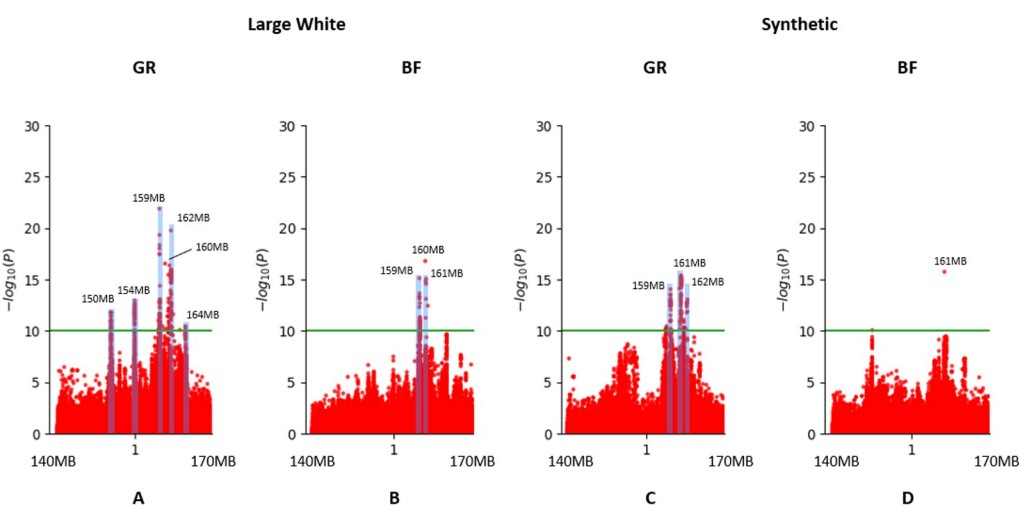

**Fig 6. Manhattan plots of region on chromosome 1 showing several QTLs for growth rate (GR) and backfat (BF).**

by the top SNP at 160Mb in Fig 5, which is in high LD with the *MC4R* missense SNP. The SNP in the QTL at 160Mb for backfat, as well as the QTLs for both traits at 160Mb in Synthetic, also show LD with the *MC4R* missense SNP.

The top SNP for growth rate in Large White for this whole region is located at 159,788,889bp with a MAF of 10% and a negative effect on growth. This SNP and the variants that are in LD are all intergenic variants, located near the *CDH20* (cadherin 20) gene. We also find significant intron variants of *CDH20*, however, some of these SNPs have a frequency of 23% and a positive effect on growth, while others have a frequency of 3% and a negative effect on growth. The top SNP of 23% MAF is located at 159,869,511bp and the top SNP of 3% MAF is located at 159,821,786bp. The same locus around the *CDH20* gene shows up in the Synthetic breed.

We looked at gene expression in this region in liver, muscle, spleen and lung. We found several SNPs between 159 and 161Mb to significantly affect the expression of the *PYGL* (glycogen phosphorylase L) gene, located at 180Mb on the same chromosome 1, an allosteric enzyme that catalyzes the rate-limiting step in glycogen catabolism [17].

**Table 2. LD within the QTL rich 15MB region on chromosome 1 in Large White.**

| LD (R²) / SNP | 150364747 | 150364747 | | | | | | | | |
|---|---|---|---|---|---|---|---|---|---|---|
| 150364747 | 1 | **154978577** | | | | | | | | |
| 154978577 | 0.216699 | 1 | **159788889** | | | | | | | |
| 159788889 | 0.207151 | 0.0420421 | 1 | **159821786** | | | | | | |
| 159821786 | 0.0492199 | 0.00678435 | 0.280964 | 1 | **159869511** | | | | | |
| 159869511 | 0.104494 | 0.495845 | 0.0342459 | 0.00947161 | 1 | **160883673** | | | | |
| 160883673 | 0.166053 | 0.696078 | 0.0550575 | 0.0154966 | 0.48608 | 1 | **162020747** | | | |
| 162020747 | 0.194158 | 0.0348452 | 0.726328 | 0.237485 | 0.0274089 | 0.0538396 | 1 | **162062651** | | |
| 162062651 | 0.0639927 | 0.00931752 | 0.202506 | 0.677308 | 0.0112798 | 0.0187777 | 0.339906 | 1 | **164838503** | |
| 164838503 | 0.0195427 | 0.157152 | 0.0581801 | 0.00639335 | 0.0786844 | 0.219678 | 0.042347 | 0.00143272 | 1 | **164909824** |
| 164909824 | 0.0425097 | 0.00175518 | 0.147268 | 0.00425724 | 0.000552646 | 0.00572898 | 0.131389 | 0.0052275 | 0.146082 | 1 |

LD, Linkage Disequilibrium; Cell shading; orange: R² = 0.40–0.60, yellow: R² = 0.60–0.70, green: R² > 0.70

## Discussion

In this study, we performed sequence based GWAS on a large scale with both an additive and non-additive model, allowing us to identify novel and low frequency deleterious variants. By using a non-additive (recessive) model, we identified completely novel QTLs compared to the additive model, which are in most cases extremely significant due to their deleterious nature. As these recessive deleterious variants tend to be present with low allele frequencies, it is essential to have large datasets when attempting to identify them through GWAS. Especially when using WGS data, this leads to very computationally intense analyses. We experienced mainly the high number of phenotypes to be a bottleneck for memory usage, as the genome can be split up in segments to run parallel analysis. The computational limitations are naturally very dependent on the type of software used and the available computing infrastructure. In our situation, we found a number of around 75,000 phenotypes to be the maximum. This number is sufficient when mainly focusing on SNPs with MAFs above 1%, which was the case in this study due to imputation accuracies. If in the future we would be able to accurately impute even lower frequency SNPs with high accuracy, new methods would be needed to include a larger number of phenotypes for GWAS.

We observed a total of 23 QTLs across the two traits and breeds, of which most will require further research to proof the causal variants and the biological mechanism. However, for some of the QTLs we identified potential causal variants causing a loss of function and therefore expected to impair gene function. We found some of these variants to cause very poor performance and therefore proving very interesting candidates for selection.

One of these variants is located on chromosome 2, where we observed a QTL in the non-additive GWAS, leading to a reduction in both growth rate and backfat. The most significant SNPs in this QTL are mainly intron variants of *ADAMTS2*. We do not observe any deleterious variants in this gene specifically. Inactivation of *ADAMTS2* has been associated with Ehlers-Danlos syndrome, a disorder affecting collagen formation and function [18]. Affected individuals suffer from hypermobile joints and flexible, fragile skin [19]. There are several mutations known in several genes that can lead to development of this disorder. Though not a very common symptom, short stature has been observed in some patients with Ehlers-Danlos syndrome [20]. The syndrome has been observed in domestic animals including sheep and cattle, but not yet in pigs. Affected animals also display the loose and fragile skin phenotype [21]. If the decreased growth we observe in pigs homozygous for this region were to be caused by a similar syndrome, we would expect to observe similar phenotypes. This would require further observation of homozygous animals. For now, since we do not observe deleterious variants in *ADAMTS2* and are not aware of additional symptoms indicating Ehler-Danlos syndrome in these animals, we do not consider this disorder to likely be the cause of the reduced growth. Another significant SNP in this QTL, though not as significant as the top SNPs, is a missense SNP in *CDHR2* with a very strong deleterious effect. Previous research has studied the function of *CDHR2* by knocking out the gene in mice. Knock-out animals showed a decrease in body weight, likely due to absence of *CDHR2* leading to shorter microvilli and a lower packing density in the intestine [14]. Based on these findings we expect this missense SNP could be the causal variant leading to the observed decrease in growth and backfat.

Another recessive causal loss of function variant we managed to identify is located on chromosome 15, affecting the *OBSL1* gene. This variant causes the strongest phenotypic effect we observed in all GWAS results. Homozygous animals show a strong reduction in growth rate and increase in backfat thickness. We identified a frameshift variant in *OBSL1* to be the causal mutation. Cytoskeletal adaptors play an important role in ensuring structural integrity of cells by linking the internal cytoskeleton to the cells membrane [22]. Defects in *OBSL1* have been

found to lead to 3M-syndrome 2 in humans. Several cases have been studied and all mutations in *OBSL1* causing the syndrome were null mutations within the first 6 exons of the gene, therefore affecting all transcripts [23]. The variant we identified is located on exon 5 and causes a frameshift, therefore likely also leading to the gene being fully defective. The most common symptom of 3M-syndrome is short stature due to growth restriction, often accompanied by dysmorphic features and skeletal abnormalities [24]. Similar symptoms have been observed in sheep with a defect in *OBSL1*, though unlike in humans, sheep homozygous for the defect are stillborn [25]. 3M-syndrome has not yet been described in pigs before. We expect the identified frameshift variant to cause a similar syndrome in pigs, explaining the severe reduction in growth we observe in homozygous animals. We attempted to obtain homozygous individuals, but there were no current ongoing carrier by carrier matings expected to produce homozygous offspring.

On chromosome 7 we found a deleterious missense SNP in the *MPIG6B* gene in the Synthetic breed, which hasn't been directly associated with variations in growth/body size. However, this gene is essential for blood platelet production and function.

Previous research has shown that knock-out of the gene in mice leads to a reduced number of blood platelets and the occurrence of enlarged platelets. Additionally, these animals showed increased production of metalloproteinase, leading to increased shedding of cell-surface receptors [26]. We speculate that the missense SNP we identified might lead to similar issues in the blood of homozygous animals, expressing through overall lower performance including decreased growth.

Deleterious recessive variants are often observed segregating in a single breed [2]. Therefore, crossbred offspring are generally not affected by these variants, which has been hypothesized to contribute to heterosis [27]. We did identify one recessive stop-gain SNP present in both lines, in the *ANKRD55* gene. The presence of this SNP can now be monitored and crossbred mating resulting in potential homozygous offspring can be avoided. Variants with large effects found only in a single line can be monitored and prevented from segregating further within the line.

We also found some interesting novel QTLs in our additive GWAS for which we have not yet managed to pin down specific causal variants, but we identified genes that are likely involved. At the start of chromosome 2, we found a highly significant QTL with a frequency of only 0.5%, leading to strongly increased backfat. In this QTL, both the *INS* and *IGF2* genes are present. *IGF2* produces a protein important in growth regulation and is also involved in glucose metabolism [28], and *INS* plays a role in carbohydrate and lipid metabolism by regulating glucose uptake [29]. A QTL in *IGF2* has been previously identified to affect fat deposition in pigs [30]. Therefore, we expect *IGF2* to most likely be the causal gene, but we have not yet found a specific variant linked to it. Finding the causal variant is especially hard due to the extremely low frequency of the QTL.

One of the most difficult to analyze results is the small complex region on chromosome 1 with several significant effects on growth rate and backfat. The regions with different effects overlapping with each other makes it very challenging to identify causal variants for these QTLs. We did find several SNPs for both breeds that have a significant effect on the expression of *PYGL*. *PYGL* functions to create an enzyme that breaks down glycogen into glucose in the liver [31]. Glycogen levels influence fat metabolism [32], explaining how changes in expression of this gene could lead to reduced or increased growth rate and backfat thickness. In Large White, the SNPs affecting *PYGL* expression are associated with a positive effect on growth, whereas in Synthetic they are associated with a negative effect on growth. This likely indicates that it's not the same causal variant affecting *PYGL* expression in both breeds. Further in-depth investigation of this region will be needed to disentangle all separate variants and effects

in this QTL. This region highlights the benefit of using a high-resolution data set, as previous analysis often considered this QTL to be caused only by the *MC4R* missense variant, whereas we can show that several other variants and genes are involved.

The *CDH20* gene present in the chromosome 1 locus has been previously associated with fat deposition [33]. Additionally, this study reports the *KIAA1549* gene, which we also observed in the QTL on chromosome 18 affecting backfat thickness. Another recent study has associated the *ANO9* gene (chromosome 2) with increased backfat thickness [34]. We identified an additive missense SNP in this gene associated with very high backfat thickness, however we consider *IGF2* and *INS* to be more likely candidate genes to cause this effect.

In conclusion, by performing a large-scale sequence based GWAS using a non-additive model we identified several rare, recessive, and deleterious variants with high impact on pig performance and production. Additionally, the high-resolution capacity of this data set enabled us to detect multiple independent QTL effects in the well-known *MC4R* region. These results provide a valuable resource for breeding and for further reduction of the frequency of deleterious alleles.

## Methods

### Ethics statement

Samples collected for DNA extraction were only used for routine diagnostic purpose of the breeding programs, and not specifically for the purpose of this project. Therefore, approval of an ethics committee was not mandatory. Sample collection and data recording were conducted strictly according to the Dutch law on animal protection and welfare (Gezondheids- en welzijnswet voor dieren).

### Genotyping & sequencing

All animals imputed to sequence were initially genotyped on either (Illumina) Geneseek custom 50K or 25K SNP chips, with 50,689 SNPs and 26,894 SNPs respectively (Lincoln, NE, USA). The chromosomal positions were determined based on the Sscrofa11.1 reference assembly [35]. SNPs located on autosomal chromosomes were kept for further analysis. Next, the SNPs were filtered using the following requirements: Each marker had a MAF greater than 0.01, and a call rate greater than 0.85, and each animal a call rate greater than 0.7. SNPs with a p-value below 1E-5 for the Hardy-Weinberg equilibrium exact test were also discarded. All pre-processing steps were performed using Plink v1.90b3.30 [36]. The reference population for imputation to 660K was genotyped on the Axiom porcine 660K SNP array from Affymetrix. Quality control was as described above for the 50K genotyping.

DNA sequencing of the reference population was performed on the Illumina Hiseq. The average read length was 150bp, the average coverage was 14.6X and the average mapping quality was 37. The reads were aligned to Sus Scrofa 11.1 [35] using BWA-MEM v0.7.17 [37]. Variant calling was performed with Freebayes v1.3.1 with settings—min-base-quality 20,—min-mapping-quality 30,—min-alternate-fraction 0.2,—haplotype-length 0 and—min-alternate-count 3 [38]. Variants with a quality score below 20 were discarded. Variants were annotated using the Ensembl variant effect predictor (VEP, release 103) [11].

### Imputation

For imputation from 50K to 660K density we used Fimpute v3.0 [39]. The reference population consisted of 3500 animals of different breeds.

The first step in imputing to sequence is phasing of the haplotypes. For the phasing we used Beagle 5.4, with a window of 20, overlap of 5, Ne of 100 and 16 threads [40].

We then ran the conform-gt tool to get consistent allele coding between the reference and target VCF files. For the actual imputation we used Beagle 5.4, with a window of 3, Ne of 100 and 20 threads [4]. For Large White, one round of imputation to sequence was performed for 40,000 animals, and another round for 80,000 animals. We then merged the resulting VCF files of both imputation runs into one file for each chromosome using bcftools merge [41], giving us sequence data on a total of 117,244 animals after some were lost in the phasing steps.

For Synthetic we performed imputation to sequence on 80,000 animals. We used Plink v1.90b3.38 [36] to recode the VCF files so that all major alleles were set as reference alleles. To obtain more information on each SNP we used the bcftools fill-tags plugin [41]. The reference population consisted of 884 whole genome sequences for the imputation of the first 40,000 Large White animals, and 1069 whole genome sequences for the second imputation of 80,000 animals as well as the Synthetic animals. This reference population contained animals of the Large White, Synthetic, Landrace, Duroc and Pietrain breeds.

Imputation accuracies were evaluated based on the $R^2$ values given by Beagle for each variant, which indicated the squared correlation between the true number of non-major alleles on a haplotype and the posterior imputed allele probability [4].

### Genome-Wide Association Studies (GWAS)

We performed single-SNP GWAS on the imputed sequence data using GCTA v1.93.2 [42] with the following linear model:

$$y*k = \mu + X\hat{\beta} + uk + ek$$

where $y^*k$ is the pre-corrected phenotype of the $k$ animal (pre-corrected for all non-genetic effects); μ is the average of the pre-corrected phenotype; $X$ is the genotype, coded as 0, 1, or 2 copies of one of the alleles of the $k$ animal for the evaluated SNP; $\hat{\beta}$ is the unknown allele substitution effect of the evaluated SNP; $u_k$ is the residual polygenic effect, assuming u $\sim$ N (**0, G** $\sigma2u$), which accounted for the (co)variances between animals due to relationships by formation of a **G** matrix (genomic numerator relationship matrix build using the imputed genotypes), $\sigma2u$ is the additive genetic variance; and ek is the random residual effect which was assumed to be distributed as $\sim$ N (**0, I** $\sigma2e$).

To run the non-additive model GWAS, all heterozygote genotypes were set to 0/0 to test the phenotypes of wildtype (homozygous for major allele) and heterozygous animals against homozygous (for minor allele) animals.

### Gene expression

We had access to a gene expression dataset including expression data from 100 crossbred animals in four tissues: liver, spleen, lung and muscle [43].

### Phenotypes

The phenotypes were pre-corrected for non-genetic effects. For the Large White breed, a sow line, we used 67,280 growth rate phenotypes and 72,061 phenotypes for backfat thickness from animals born from 2012 to 2022. For the Synthetic breed, a boar line, we used 74,145 phenotypes for both traits from animals born from 2017 to 2022.

### Validation of causal 5 bp *OBSL1* Deletion

PCR was done using 60 ng of genomic DNA, with 0.4 μm of each primer, 1.8 mM MgCl2, and 25 units/ml OneTaq DNA Polymerase (OneTaq 2X Master Mix with Standard Buffer, New

England Biolabs) in manufacturer's PCR buffer in a final volume of 12 µl. Initial denaturation for 1 min at 95˚C was followed by 35 cycles of 95˚C for 30 s, 55˚C for 45 s, 72˚C 90 s, followed by a 5 min extension 72˚C. PCR primers for OBSL1 are ACGTCCTTGATCCTGTCTGC forward and CTCTCCACCATCATCCAGGG reverse. The forward primer was dye-labeled with 6-FAM to produce a fluorescently labeled PCR product detectable on ABI 3730 DNA sequencer (Applied Biosystems). Fragment sizes were determined using GeneMapper software 5 from ABI.

### Further analysis & figures

To perform linkage disequilibrium (LD) analysis we used Plink v1.90b3.38 with settings—ld-window-r2 0—ld-window 99999999—ld-window-kb 100000 [36].

To assess how deleterious a variant is we considered the SIFT score as given by the VEP [11], as well as the Combined Annotation Dependent Depletion (CADD) score [44], adapted for pigs (pCADD, [13]). Boxplots, heatmaps and manhattanplots were made using the python packages seaborne [45], heatmapz [46] and QMplot [47] respectively. Additionally, pandas [48] and matplotlib [49] were used in creating the figures.

### Supporting information

**S1 Fig. Overview of the imputation and GWAS pipeline.**
(TIF)

**S2 Fig. PCA plot of GWAS populations.**
(TIF)

**S1 Table. Top SNPs for all QTLs resulting from the GWAS.**
(PDF)

**S2 Table. Validation of causal 5bp *OBSL1* deletion.**
(PDF)

### Acknowledgments

We thank Imke Leemans and Frank van Haaren for their help in farm communication and data collection.

### Author Contributions

**Conceptualization:** Anne Boshove, Martijn F. L. Derks, Barbara Harlizius.

**Formal analysis:** Anne Boshove, Martijn F. L. Derks, Claudia A. Sevillano, Maren van Son.

**Methodology:** Anne Boshove, Martijn F. L. Derks, Claudia A. Sevillano, Marcos S. Lopes.

**Resources:** Bert Dibbits.

**Validation:** Bert Dibbits.

**Visualization:** Anne Boshove.

**Writing – original draft:** Anne Boshove.

**Writing – review & editing:** Anne Boshove, Martijn F. L. Derks, Claudia A. Sevillano, Marcos S. Lopes, Maren van Son, Egbert F. Knol, Barbara Harlizius.

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
