## [Decision Letter · Decision Letter 0]

30 Nov 2023

Dear Dr Boshove,

Thank you very much for submitting your Research Article entitled 'Large scale sequence-based screen for recessive variants allows for identification and monitoring of rare deleterious variants in pigs' to PLOS Genetics.

The manuscript was fully evaluated at the editorial level and by independent peer reviewers. The reviewers appreciated the attention to an important problem, but raised some concerns about the current manuscript. Based on the reviews, we will not be able to accept this version of the manuscript, but we would be willing to review a revised version. We cannot, of course, promise publication at that time.

As you can see from the reviewers' comments the required revisions to the manuscript are not very substantial. However, there is a concern among the editors regarding the availability of the underlying raw data. Please consider to make more of the raw data publicly available (e.g. phenotypes of all animals, complete genotypes (array and imputed), accessions for the high coverage WGS data).

Should you decide to revise the manuscript for further consideration here, your revisions should address the specific points made by each reviewer and the editorial comment about data availability. We will also require a detailed list of your responses to the review comments and a description of the changes you have made in the manuscript.

If you decide to revise the manuscript for further consideration at PLOS Genetics, please aim to resubmit within the next 60 days, unless it will take extra time to address the concerns of the reviewers, in which case we would appreciate an expected resubmission date by email to plosgenetics@plos.org.

We are sorry that we cannot be more positive about your manuscript at this stage. Please do not hesitate to contact us if you have any concerns or questions.

Yours sincerely,

Tosso Leeb, PhD

Academic Editor

PLOS Genetics

Gregory Barsh

Editor-in-Chief

PLOS Genetics

Reviewer's Responses to Questions

**Comments to the Authors:**

Reviewer #1: I only have some very minor comments. This is an impressive study that addresses a difficult problem in livestock genomics and other species, capturing effects of low frequency variants.

Page 8, line112; novel low “frequency” deleterious alleles

Page 12, line 168; We “found” a very significant

Page 14, lines 213-217; could you provide the rs# for these SNP if known?

Page 14, line 218; In heading, Low “frequency” additive QTL

Page 14, line 220; These SNPs have very low “frequencies” with MAFs

Page 17, line 272; low “frequency” deleterious variants

Page 17, line 282; lower “frequency” SNPs

Page 18, line 314; gene being fully “defective”.

Page 21, lines 376-380 or page 23, lines 429-431; could you describe the animal populations in more detail?

Page 21, line 390; could you describe the sequencing in more detail, i.e., coverage, read length etc.

Page 23, line 427; do these gene expression datasets have accession numbers? Describe how SNP were associated with expression in more detail.

Page 23, line 428; dye-labeled with “either”?

Although not essential, recent reports have also identified a few of the candidate genes you report for growth and fatness QTL and could be included and discussed.

Shi L, Wang L, Fang L, Li M, Tian J, Wang L, Zhao F. Integrating genome-wide association studies and population genomics analysis reveals the genetic architecture of growth and backfat traits in pigs. Front Genet. 2022 Nov 25;13:1078696. doi: 10.3389/fgene.2022.1078696. PMID: 36506319; PMCID: PMC9732542.

Heidaritabar M, Bink MCAM, Dervishi E, Charagu P, Huisman A, Plastow GS. Genome-wide association studies for additive and dominance effects for body composition traits in commercial crossbred Piétrain pigs. J Anim Breed Genet. 2023 Jul;140(4):413-430. doi: 10.1111/jbg.12768. Epub 2023 Mar 7. PMID: 36883263.

Reviewer #2: The authors used a large-scale pig data set with in total 200k genotyped and phenotyped sows to detect recessive deleterious variants. The genomes of the individuals are imputed up to whole genome sequence-level genotypes. The applied GWAS approach is straightforward and comparable simple, but it was successfully applied and the results are very interesting and discussed in detail. The paper is very well written. In summary, I would classify it as a higher scientific content study and as such it fits well to the Journal.

I have a couple of minor remarks.

1. The data sets were analysed separately to contrast the results between the Large White and Synthetic data set. The question is why, in addition, a joint analysis was not conducted to utilize the full power to detect the recessive alleles. The joint analysis could be applied using summary data already generated within the data sets by applying a meta-analysis.

2. Line 422: ‘the non-additive model GWAS, all heterozygote genotypes were set to 0/0 to test the phenotypes of wildtype and heterozygous animals against homozygous animals‘. It is not clear how the wild type genotype was identified. In addition, why not applying an additive and a dominance effect simultaneously to the model? This would result in one extra parameter estimate and the power of the experiment should be high enough to detect the main recessive effects also under this parametrization.

3. How were the indels modeled under the applied GWAS parametrization?

4. No population stratification effects on the GWAS results are shown. I suggest to add a lambda value and / or a qq plot to show some putative inflation of significant hits.

5. Imputation accuracy is an important parameter for these kind of analysis with a focus of rare variants. How was this evaluated in detail? This becomes not clear at present.

6. I’m a bit surprised by he very clear signals spanning a low genomic region. This must have been a result from a very high mapping resolution, i.e. s strong LD decay. This is partly shown in Table 2, but a genomewide illustration 7 analysis of the LD structure would be helpful to understand the high mapping resolution.

7. A bit more explanation is needed to understand Figure 5A.

Reviewer #3: Thanks for the interesting manuscript entitled 'Large scale sequence-based screen for recessive variants allows for identification and monitoring of rare deleterious variants in pigs'. It was very well written. In following you can see my comments and questions:

1) Please explain, why you have chosen only these two traits i.e., growth rate and backfat thickness?

Why these two traits are important to find novel QTLSs? What about the other traits such as feed consumption, loin muscle thickness, intramuscular fat or the other production traits?

2) Figure 2: Am I correct? The significant SNPs for BF on chromosome 1: 224 Mb were found for synthetic breed. In figure 4, I see two blocks for this region for Large White pigs. Similar for chromosome 2: 2 MB two blocks for BF for synthetic breed. Is it not supposed to be for Large White pigs? I assume the legend are reverse. Up is for synthetic breed and down is for Large White breed.

3) Lines 315-320: Did you have some homozygous animals for this QTL on Chr15? That could the expression of OBSL1 gene is not like human or even sheep. Based on the study in sheep, lambs with hypoplasia syndrome were stillborn.

4) Line 321: Please mention the position of this SNP in the Synthtic breed (Chromosome 7: 23.8 Mb).

5) Line 331-333: Monitoring and avoidance of crossbred mating can be suggested only for this SNP?

Best regards

**Have all data underlying the figures and results presented in the manuscript been provided?**

Reviewer #1: Yes

Reviewer #2: None

Reviewer #3: Yes

PLOS authors have the option to publish the peer review history of their article (what does this mean?). If published, this will include your full peer review and any attached files.

Reviewer #1: No

Reviewer #2: No

Reviewer #3: **Yes: **Negar Khayatzadeh

---

## [Editor Report · Decision Letter 1]

27 Dec 2023

Dear Dr Boshove,

We are pleased to inform you that your manuscript entitled "Large scale sequence-based screen for recessive variants allows for identification and monitoring of rare deleterious variants in pigs" has been editorially accepted for publication in PLOS Genetics. Congratulations!

Yours sincerely,

Tosso Leeb, PhD

Academic Editor

PLOS Genetics

Gregory Barsh

Editor-in-Chief

PLOS Genetics

Comments from the reviewers (if applicable):

**Data Deposition**

http://datadryad.org/submit?journalID=pgenetics&manu=PGENETICS-D-23-01196R1

**Press Queries**

---

## [Editor Report · Acceptance letter]

4 Jan 2024

PGENETICS-D-23-01196R1 

Large scale sequence-based screen for recessive variants allows for identification and monitoring of rare deleterious variants in pigs 

Dear Dr Boshove, 

We are pleased to inform you that your manuscript entitled "Large scale sequence-based screen for recessive variants allows for identification and monitoring of rare deleterious variants in pigs" has been formally accepted for publication in PLOS Genetics! Your manuscript is now with our production department and you will be notified of the publication date in due course.

With kind regards,

Zsofi Zombor

PLOS Genetics

On behalf of:
